# Evaluating targeted COVID-19 vaccination strategies with agent-based modeling

**Thomas J. Hladish** [1,2]*, **Alexander N. Pillai** [1], **Carl A. B. Pearson** [3,4], **Kok Ben Toh** [1,5], **Andrea C. Tamayo** [1], **Arlin Stoltzfus** [6], **Ira M. Longini** [2,7]

**1** Department of Biology, University of Florida, Gainesville, Florida, United States of America, **2** Emerging Pathogens Institute, University of Florida, Gainesville, Florida, United States of America, **3** Department of Infectious Disease Epidemiology, London School of Hygiene & Tropical Medicine, London, United Kingdom, **4** South African DSI-NRF Centre of Excellence in Epidemiological Modelling and Analysis (SACEMA), Stellenbosch University, Stellenbosch, South Africa, **5** Institute of Global Health and Department of Preventive Medicine Northwestern University, Chicago, Illinois, United States of America, **6** Office of Data and Informatics, National Institute of Standards and Technology, Gaithersburg, Maryland, United States of America, **7** Department of Biostatistics, University of Florida, Gainesville, Florida, United States of America

* tjhladish@gmail.com

**Data Availability Statement:** All data and code used for running simulation experiments, model fitting, and plotting is available on a GitHub repository at https://github.com/tjhladish/covid-

## Abstract

We evaluate approaches to vaccine distribution using an agent-based model of human activity and COVID-19 transmission calibrated to detailed trends in cases, hospitalizations, deaths, seroprevalence, and vaccine breakthrough infections in Florida, USA. We compare the incremental effectiveness for four different distribution strategies at four different levels of vaccine supply, starting in late 2020 through early 2022. Our analysis indicates that the best strategy to reduce severe outcomes would be to actively target high disease-risk individuals. This was true in every scenario, although the advantage was greatest for the intermediate vaccine availability assumptions and relatively modest compared to a simple mass vaccination approach under high vaccine availability. Ring vaccination, while generally the most effective strategy for reducing infections, ultimately proved least effective at preventing deaths. We also consider using age group as a practical surrogate measure for actual disease-risk targeting; this approach also outperforms both simple mass distribution and ring vaccination. We find that quantitative effectiveness of a strategy depends on whether effectiveness is assessed after the alpha, delta, or omicron wave. However, these differences in absolute benefit for the strategies do not change the ranking of their performance at preventing severe outcomes across vaccine availability assumptions.

## Author Summary

We use our agent-based model of SARS-CoV-2 transmission to evaluate alternative vaccine distribution strategies over a range of vaccine supply scenarios. We find that strategies targeting transmission (*e.g.*, ring vaccination) perform best in preventing infections, but targeting disease risk prevents more instances of severe outcomes. Specifically, strategies based on age, or age and comorbidities—which do not require contact tracing—resulted in the fewest hospitalizations in our model. These strategy rankings held true

abm. Simulation code bundled with synthetic population input files is availble from DOI: 10.5281/zenodo.7601664. Code used to generate the synthetic population input files is available from https://github.com/kokbent/synthpop-fl.

**Funding:** This work was supported in part by The Emerson Collective (https://www.emersoncollective.com/) to IML, The Ron Conway Family to IML, NIH/NIAID (R56AI148284; https://www.niaid.nih.gov/) to IML, the UFII COVID-19 SEED Fund (https://informatics.research.ufl.edu/) to TJH, and the International Decision Support Initiative (https://www.idsihealth.org/), which is funded by the Bill and Melinda Gates Foundation (OPP1202541; https://www.gatesfoundation.org/), and the World Health Organization (2022/126532; https://www.who.int/). The funders had no role in study design, data collection and analysis, decision to publish, or preparation of the manuscript.

**Competing interests:** The authors have declared that no competing interests exist.

across all vaccine supply scenarios and were robust to the introduction of SARS-CoV-2 variants. While the quantitative results cannot be directly applied to other settings (as we used a synthetic population calibrated to the State of Florida), the rankings of strategies should be more generalizable.

## Introduction

During outbreak or pandemic situations, public health agencies respond with various interventions to contain and mitigate spread of the pathogen. When available, vaccination can be a useful strategy to reduce both transmission and infection severity. However, different deployment strategies for vaccination vary in effectiveness, and in ways that may depend on vaccine performance and the natural history of the infection [1–3]. During the COVID-19 pandemic in the United States (USA), vaccination generally proceeded via passive, mass distribution, from older to younger age groups, with priority access for *e.g.*, health care workers and individuals at high risk for severe outcomes [4]. However, vaccines can also be used in more proactive strategies, like ring vaccination for Ebola, where responders vaccinate contacts and contacts-of-contacts of identified cases [5].

During outbreaks, vaccination-as-containment strategies are often reserved for pathogens with a relatively long generation time, so that vaccination can be administered and achieve efficacy prior to exposure [6, 7]. Essentially, the vaccination effort needs to outpace the spread of the pathogen. For pathogens that spread very quickly or frequently cause infections without a known exposure, vaccination strategies aim to vaccinate the most people, often preferring those with higher risk for severe outcomes [8, 9]. In general, "passive" interventions—where officials issue guidance, and make resources publicly available—will tend to be less resource-intensive than "active" interventions that require seeking out intervention targets. Choice of strategy may also depend on vaccine performance: containment requires efficacy against transmission, whereas targeting high-risk individuals requires a vaccine that provides good protection against severe outcomes if infection occurs.

SARS-like (*e.g.*, SARS-CoV-1, MERS) outbreaks have been controllable in the past with a range of active measures [10, 11]. In some settings, SARS-CoV-2 responses have included substantial, effective contact tracing programs while infection prevalence was relatively low [12, 13]. Plausibly, active vaccination approaches would be beneficial as well. In this study, we evaluated various vaccination strategies using a detailed agent-based simulation model of human activity and SARS-CoV-2 transmission. With reference to a counterfactual simulation where vaccines are distributed passively (*i.e.*, via a proportional-allocation mass campaign), we evaluated the effectiveness of active strategies, including quarantining, ring vaccination, age-prioritized vaccination, and true risk-prioritized vaccination. Many locales have prioritized vaccination by age, as a proxy for knowing actual, individual-specific risk of hospitalization (or death) given infection. As our simulated risk-prioritization strategy has this information, we consider it to be an upper bound on the performance of this kind of strategy.

We further evaluated whether strategy ranking is sensitive to vaccine supply by considering four levels of availability, including three chosen based on averages for low-, middle- or high-income countries world-wide, hereafter low, middle, and high supply (LS, MS, and HS), respectively. We also evaluated strategies based on data specifically for the USA, which had particularly fast early uptake of the vaccine, followed by slower-than-average (among high income countries) uptake during the second half of 2021 [14].

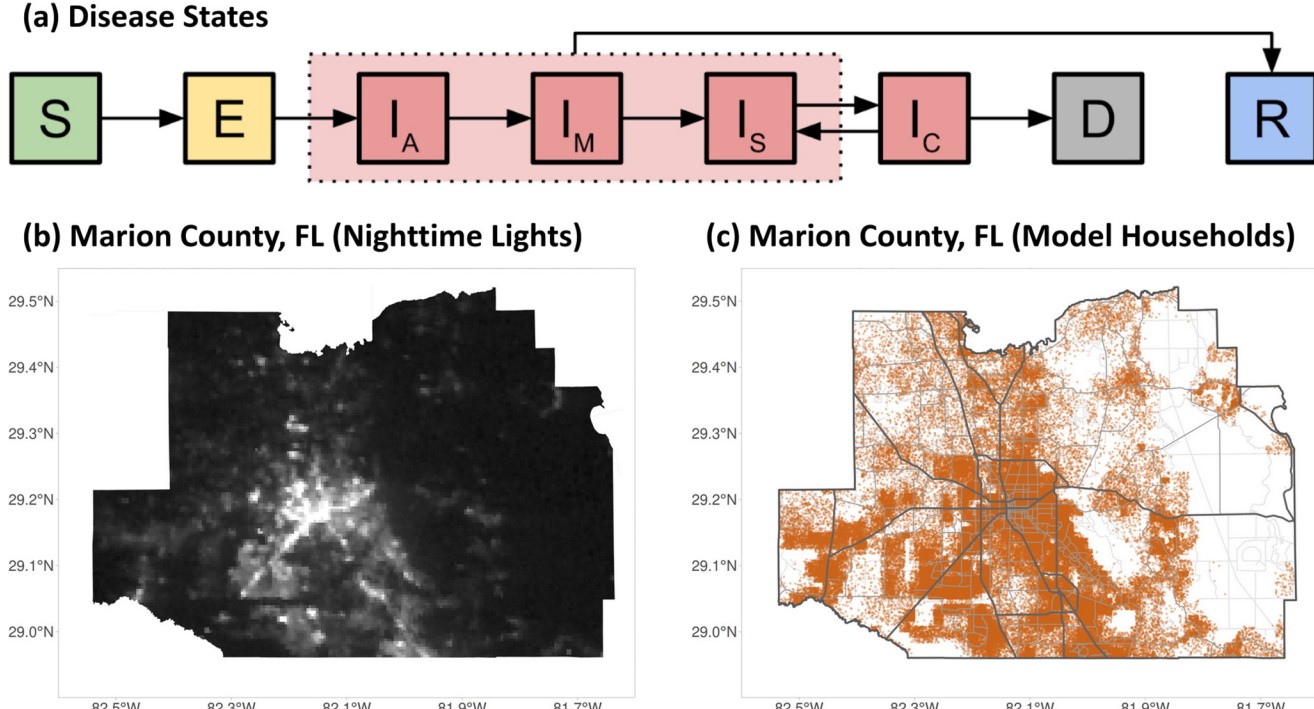

**Fig 1. Model disease states and spatial structure.** (a) Progression of the disease states in the model: susceptible (*S*) individuals may become exposed (*E*) to the virus, then progress to being infectious (initially asymptomatic [$I_A$], possibly progressing to mild [$I_M$], severe [$I_S$] or critical [$I_C$]), and finally recovering (*R*) or dying (*D*). $I_C$ individuals can either die or revert to $I_S$, in which case they will eventually recover. Recovered individuals have strain-specific immunity that changes over time. (b) Nighttime lights satellite image [16, 17] of Marion County, FL, the region used for the model's spatial structure. (c) Locations of the 161.5K model households (orange dots). Roads [18] are shown for reference but are not modeled. Nighttime lights data is available from https://eogdata.mines.edu/nighttime_light/monthly/v10/2024/202401/vcmcfg/ under the Creative Commons Attribution 4.0 International License https://eogdata.mines.edu/files/EOG_products_CC_License.pdf. Map base layer data is available from https://www.openstreetmap.org under the Open Data Commons Open Database License https://opendatacommons.org/licenses/odbl/, copyright by OpenStreetMap contributors.

## Materials and methods

We extended an agent-based model framework to support evaluating the COVID-19 epidemic and response efforts (Fig 1), previously used in [15] and derived from [1]. For a detailed description of the model, see S1 Text. The model represents SARS-CoV-2 natural history with empirically-based mechanisms in a stochastic, discrete-time transmission simulation of people moving between places (Fig 2 shows interactions; Fig 3a–3d shows time-varying model inputs). For this analysis, we consider a population of 375K people with demographics representative of Florida (and spatial structure corresponding to Marion County, Florida) as a practical sub-population with both urban and rural areas. We sample the 161.5K households in the model from state-wide microcensus data, with features including household size, ages in years, and employment or student status, and validate the model against state-wide data (Fig 3e–3i)). The presence of comorbidities relevant to COVID-19 is known in aggregate but not at the household level, and thus is sampled independently. Locations in the model (aside from residences) are based on their actual addresses in Marion County, including 45 long-term care facilities, 8.9K workplaces, 115 schools, and 6 hospitals.

People in the model engage in various interactions that may result in transmission and which are moderated by individual behavior thresholds (Fig 2). Some interactions always occur, like those within households, hospitals, and long-term care facilities. We model two population-wide non-pharmaceutical interventions (NPIs): "lockdown", where only essential

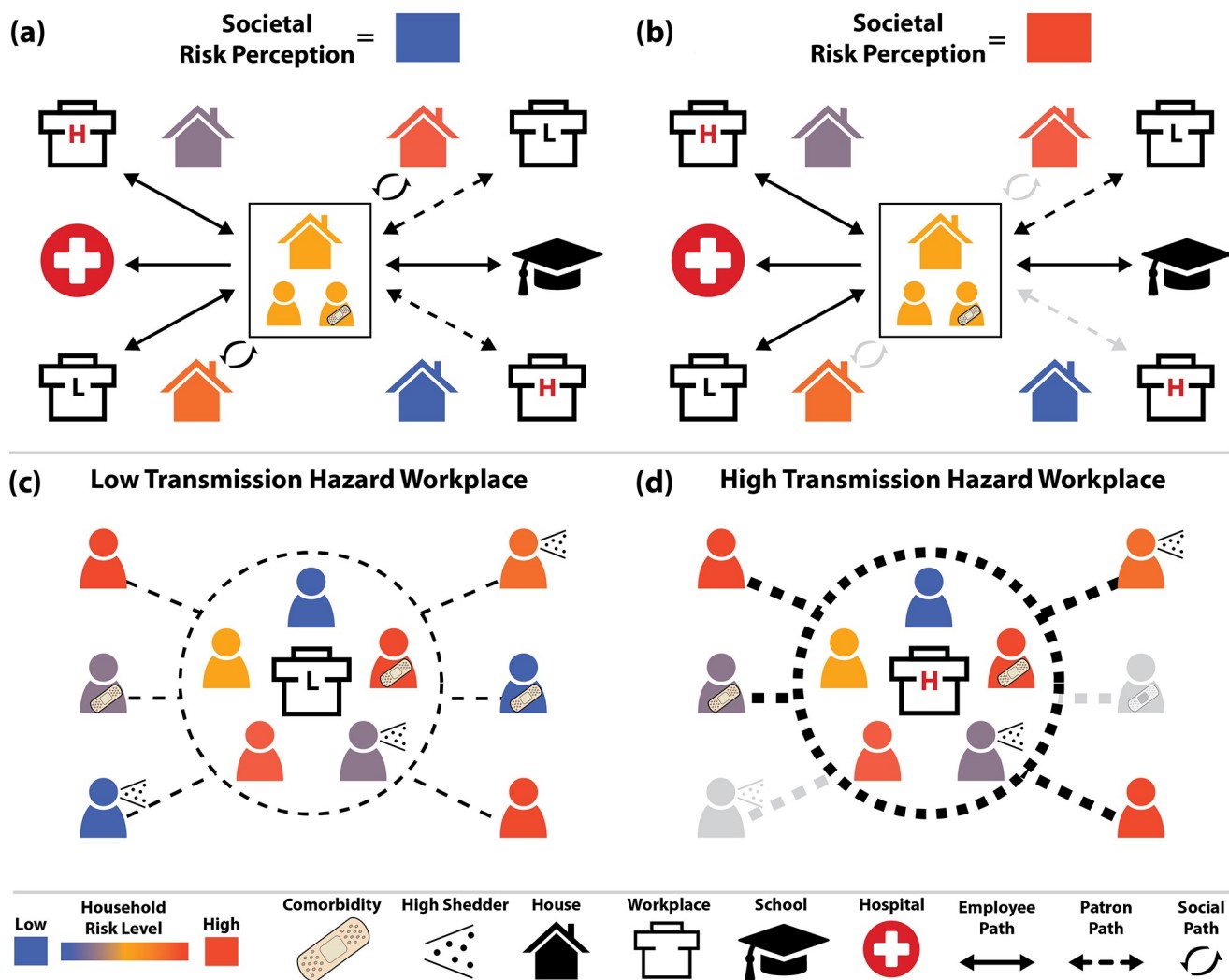

**Fig 2. Individual interactions and behaviors in the model.** Interactions occur when people in the model are in the same location at the same time, and may occur in households, workplaces, schools, hospitals, and long-term care facilities (not shown). Households have an inherent risk tolerance (indicated by color), and probabilistically have inter-household connections homophilously based on that tolerance. The population overall has a time-varying perception of risk of COVID-19 infection that may be different from the actual risk. (a) When the societal perception of risk is lower than a household's risk tolerance, household members engage in all their normal activities, including socializing with specific other households and patronizing specific high-transmission-risk workplaces like restaurants and bars. (b) When the societal perception of risk exceeds a given household's risk tolerance, the household will cease high-risk activities (indicated with greyed arrows), while maintaining more essential activities like going to work, school, and patronizing low risk workplaces (*e.g.*, grocery stores). (c, d) Employees and patrons interact in some workplace types, with interactions between employees more likely to result in transmission. (d) When perceived risk is high, risk-intolerant (blue) employees of high-transmission-hazard workplaces still go to work, while risk-intolerant patrons stop visiting these locations (and thus are grayed-out).

workplaces are open, and reduced activity in schools. These have specific periods and levels; see Section E in S1 Text. These NPIs apply in both the validation and scenario analysis simulations.

We also model personal protective behaviors (PPBs) which potentially moderate social interactions and patronage of businesses. PPBs are represented as individuals choosing not to visit social contacts and patronize high-transmission-hazard businesses if the societal risk perception exceeds an individual's risk tolerance. Risk tolerance is static and is defined at the household scale, whereas societal risk perception varies by day but is the same for all households. See Section E in S1 Text. for more details.

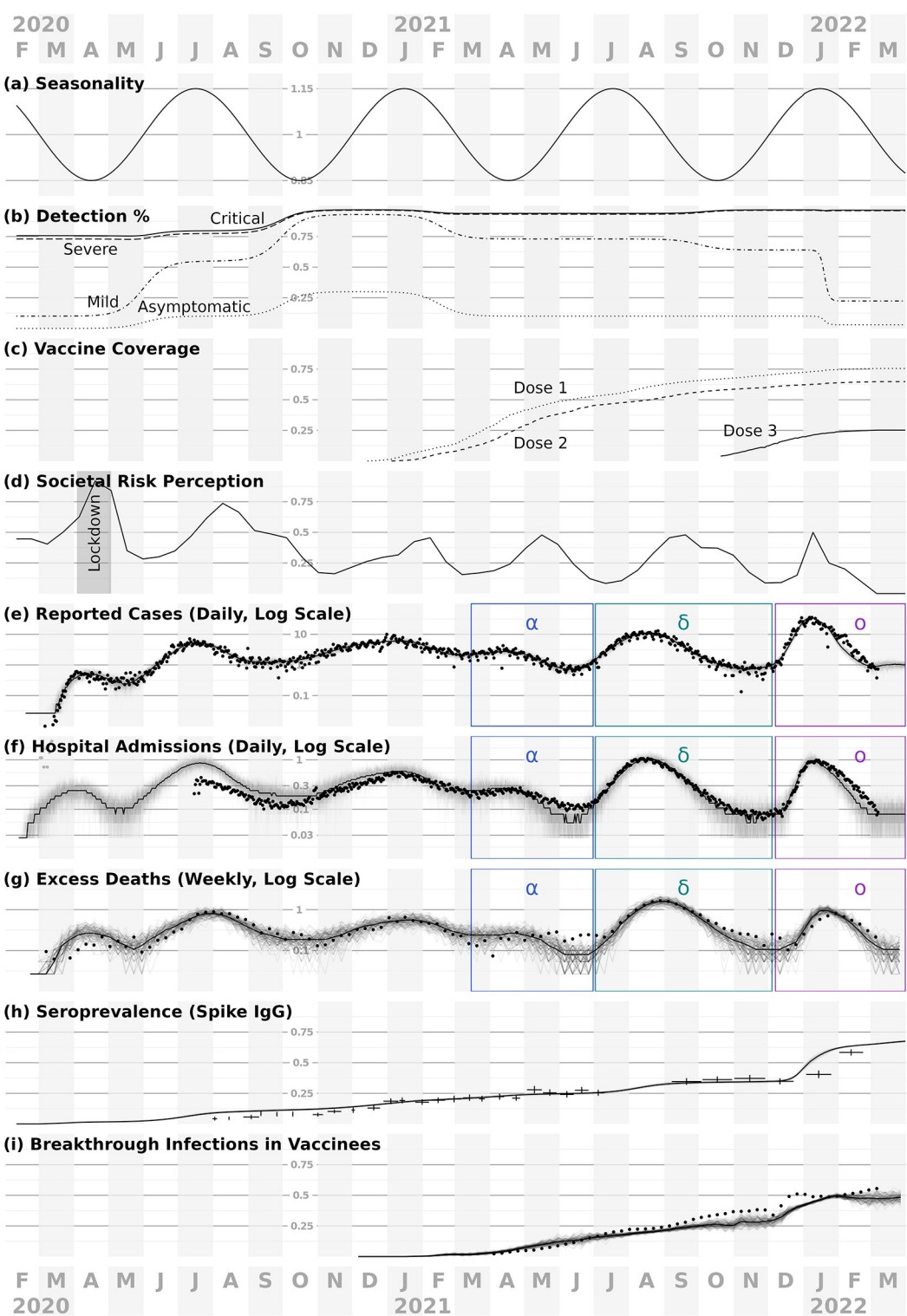

**Fig 3. Time-varying model inputs and indicators of model performance.** Panels (a-d) show model inputs, and (e-i) compare model outputs to observed data. In (e-i), points and cross-hairs indicate observed values, solid lines median trends, and faded lines sample trajectories. Horizontal gridline values are plotted above October 2020. (a) Seasonal forcing has a 6-month period, peaking in January and July each year; we also considered an alternative model with no seasonal forcing, see Section E in S1 Text. for details. (b) Detection and reporting probabilities by disease outcome. (c) Simulated first, second and third vaccine doses distributed statewide in Florida, used to calibrate the model (but not for evaluating

strategies). (d) Societal risk perception, which drives personal protective behaviors in the model, is fitted so that cumulative reported cases in the model match empirical case data for FL (black dots in panel e). For approximately the month of April 2020, non-essential businesses were closed in the state, and thus are closed during this period in the model (gray "lockdown" shaded region). Not shown: schools in the model close during the summers and during spring 2020, and are 50% and 80% open during the 2020–2021 and 2021–2022 school years, respectively. (e-i) Simulated data closely track empirical data for incidence of reported cases (e), daily hospital admissions (f), excess deaths (g), seroprevalence (h), and the fraction of infections that occurred in vaccinees (i). Results in (e-g) are scaled to show values per 10,000 individuals, and VOC waves are labeled as alpha ($\alpha$), delta ($\delta$) and omicron ($o$). For empirical seroprevalance data in (h), horizontal bars indicate the dates covered by each data point and vertical lines indicate the 95% CI).

We represent the natural history of infection with exposed and infectious states, followed by a recovered state with and strain-specific immune memory (Fig 1a). Individuals vary in the efficacy of their immune responses to infection. Exposure and infection outcomes in the model are affected by age and immune history (see Section B in S1 Text. for more details), and infection outcome is additionally affected by the presence of comorbidities.

The simulation advances in daily increments, with a series of transmission opportunities among individuals when they are co-located during their daily activity pattern (Fig 2). Age- and spatially-structured interaction patterns can emerge because activities vary by age and day of week, *e.g.*, children attend specific schools, where they interact with other children from the model's catchment area associated with that school, and individuals have specific businesses that they randomly patronize. For social interactions, households are more likely to be connected to other households with similar risk tolerance thresholds, thus infection risk-groups can emerge (see [19] for more details on this aspect of our model).

## Scenarios

We define model scenarios by three factors: (1) vaccine supply level (four levels), (2) vaccination distribution strategy (four strategies), and (3) quarantine policy (two alternatives). S2 Text also covers a fourth dimension: whether eligibility to receive vaccine is conditional on not having a known prior infection, but the main text results only consider unconditional eligibility. The four supply levels correspond to the averages for low-, middle-, and high-income countries, (LS, MS, and HS, respectively), and the USA (Fig 4). We express supply levels as available doses per 10K eligible people, and derive them from UNICEF estimates of vaccine deliveries

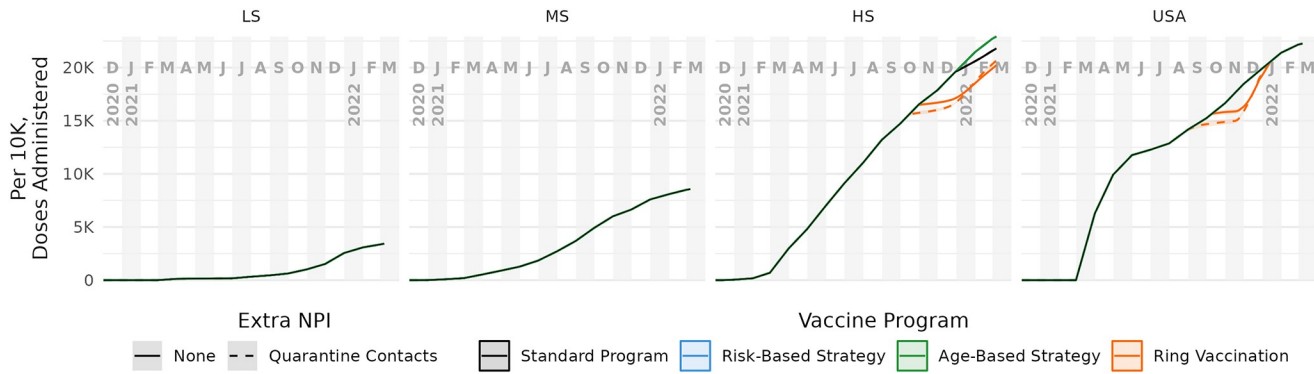

**Fig 4. Cumulative vaccine doses administered per 10k over time, by supply level and distribution strategy.** For each combination of the four supply levels (columns) and quarantine policy (dashed lines), we considered four vaccination strategies: ring vaccination (*i.e.*, infection-risk prioritization) (orange), risk prioritization (blue), age prioritization (green), and a standard mass vaccination (black). For low (LS) and middle (MS) supply levels, all strategies use all available doses. For high (HS) and USA supply levels, the strategies sometimes differ in doses delivered due to shortages of individuals eligible for revaccination; only risk- and age-based strategies always use all available doses.

[14] and WorldPop program estimates of population sizes [20]. Doses administered on a given day are the lesser of doses available and number of eligible vaccinees under the vaccination strategy. Available doses that are not consumed are rolled over to the next day. As the age- and risk-based strategies never run out of eligible vaccinees, doses administered for those strategies match doses delivered as specified by UNICEF. The supply is distributed according to a vaccination strategy, either a passive mass vaccination "standard" program (*i.e.*, completely random distribution) that we use as a baseline reference, or one of three active strategies, called "ring" (vaccinating some primary contacts of contact-traced cases, and some secondary contacts of contacts), "risk" (prioritized distribution by disease-risk deciles), or "age" (prioritized distribution by age-group deciles).

Both quarantine and ring vaccination in the model rely on contact tracing. The contact tracing is intentionally imperfect: a majority (but not all) of reported cases are traced, and the number of contacts that are identified is dependent on the setting for that contact, *e.g.*, all household members are identified, but a Poisson-distributed number of inter-household contacts will be identified. This process is repeated again for all of the identified contacts of the index case, thus both contacts and contacts-of-contacts may be identified; see Section E in S1 Text. for details. We make simplifying assumptions that contact tracing capacity is unlimited, and that it can be completed the same day an index infection is detected, which could be days or weeks after infection occurred if *e.g.* detection occurs upon hospitalization or death. Real-world contact tracing will vary substantially between settings in capacity, speed, and thoroughness, but our assumptions offer a plausible upper bound on the effectiveness of quarantine and ring vaccination strategies.

A vaccination strategy can be thought of as a way to assign individuals to a queue to receive available vaccine doses. In the (passive) standard program, individuals are assigned at random to the queue. In the ring vaccination strategy, individuals are prioritized for vaccination if they are a contact-traced primary or secondary contact of an identified case. In the risk-based vaccination strategy, the position in the queue to receive vaccine is based on risk of severe disease, calculated using comorbid status and age, and quantized by decile. A cruder but more practical strategy that has often been used for COVID-19 is to assign risk based on age alone. We consider all combinations of vaccine supply and vaccination strategy factors with and without quarantine of identified cases, primary contacts, and secondary contacts. In Fig D in S2 Text, we also report the dynamics of a no-vaccine scenario, but do not use those for reference comparison in the main text results, as COVID-19 policy challenges have generally focused on how to use the doses available, and not whether to use them.

In all cases, we assume a three-dose vaccine regimen, with an mRNA-vaccine-like efficacy profile. This approximation is a simplifying assumption to reduce the complexity of the model and translation of empirical vaccine dose data into a model input. While many countries initially used nominally single dose vaccines, those products are generally lower efficacy [21] and many were ultimately deployed with additional booster doses, making their efficacy and dose requirements more like the mRNA products (see, *e.g.*, [22]).

See Section F in S1 Text. for details on determining the dose availability and distribution time series, and Figs A and B in S2 Text for simulated distribution stratified by dose ordinality and vaccinee age group.

To compare scenarios, we simulate $N = 1000$ replicates for each scenario, with random number generator seeds matched across scenarios. This ensures identical pre-vaccination-era histories when comparing across different scenarios. We compare matched replicates by calculating cumulative outcomes (infections and deaths) by incidence, incidence averted (compared to the reference program), and relative incidence averted (*i.e.*, effectiveness) to date, then compute quantiles on these values. In S2 Text, we provide similar measures for additional

scenarios. To calculate cumulative effectiveness for an outcome $O$ (*e.g.*, infections), we take the cumulative incidence of that outcome at time $t$, for scenario $j$, $O_{j,t}$ and compare it to matching reference scenario $r$ (*i.e.*, standard distribution for the same supply level), $O_{r,t}$ as:

$$\text{CE}_{j,t} = 1 - \frac{O_{j,t}}{O_{r,t}} \qquad (1)$$

## Vaccine efficacies and dosing

Rather than simulate any particular SARS-CoV-2 vaccine product, we use published efficacy data [21] for different SARS-CoV-2 mRNA vaccines to approximate a generalized mRNA vaccine. We represent effects of vaccination using the efficacy parameters in Table F in S1 Text, namely $\text{VE}_S$ (efficacy against infection), $\text{VE}_P$ (efficacy against disease given infection), $\text{VE}_H$ (efficacy against hospitalization given disease), and $\text{VE}_I$ (efficacy against onward transmission given infection). Vaccine efficacy begins 10 days after vaccination.

Note that the typical parameter reported from clinical trials is the unconditioned efficacy against disease, $\text{VE}_{SP}$, which is related to $\text{VE}_S$ and $\text{VE}_P$ by Eq 2.

$$\text{VE}_{SP} = 1 - (1 - \text{VE}_S)(1 - \text{VE}_P) \qquad (2)$$

Each vaccination strategy determines how people become eligible for a first dose. On each day during a vaccination campaign vaccinees will be randomly drawn from the pool of all eligible people for each dose in the vaccination series. Vaccinees become eligible for second doses 21 days after the first dose, and eligible for third doses after a further 240 days (based on the median lag between second and third dose administration in Florida). Vaccinations on a given day continue until either no more doses are available, or no one is eligible.

## Results

### Model calibration

We calibrated the model through an iterative process of algorithmic fitting and manual parameters adjustment. See Section G in S1 Text. for details.

For a Florida-like population, we obtain reasonable matches to a range of metrics: observed values are mostly fluctuating about the central model trajectory and within the range suggested by the replicate trajectories (Fig 3). We use age- and dose-stratified vaccination data specific to Florida, USA for model calibration.

### Scenario analysis

We used a detailed agent-based model to compare across different scenarios, considering effects of vaccine supply levels, vaccination strategies, and quarantine policy. Epidemic curves showing simulated outcomes over time illustrate gross features such as waves due to variants that emerge at specific time-points—alpha ($\alpha$), delta ($\delta$) and then omicron ($o$)—and also more subtle effects that arise from heterogeneities such as age-structuring with respect to comorbidities, employment-based interactions, and risk of death.

The simulations begin with the approximate start of the pandemic in most countries (early 2020); as we focus on the performance of vaccination campaigns, however, we show results only for the vaccination era in the model, December 2021 to March 2022. All scenarios give identical outcomes prior to the start of interventions. Fig 5 shows cumulative infections and deaths per 10K people during the vaccination era for four supply levels. Results for severe and

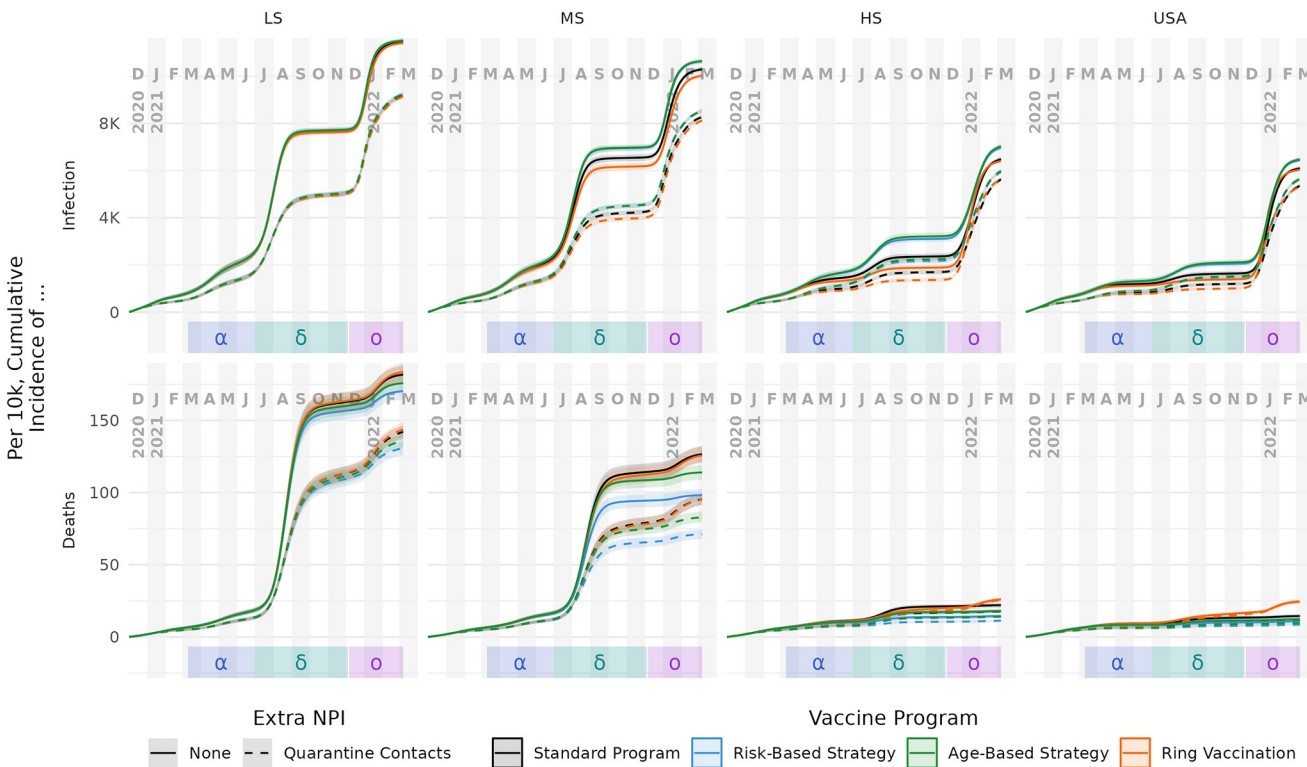

**Fig 5. Cumulative incidence of infection and death per 10k people, by supply level and distribution strategy.** Columns represent vaccine supply scenarios. Rows represent infection (top) and death (bottom) outcomes. Median values and 90% interquartile range are shown as bold lines and shaded ribbon, respectively. For infections, the major effects are supply level (columns) and the policy of quarantining (dashed lines) or not quarantining (solid lines), whereas the four vaccination strategies perform similarly. For deaths, supply level and quarantine are again the strongest factors. However, a strong effect of vaccination strategy also emerges: relative to a standard vaccine roll-out (black), risk-based vaccination (blue) and age-based vaccination (green) are more effective at preventing deaths, whereas ring vaccination (orange) is less effective. See the text for further explanation.

critical disease have the same trends as those for deaths and are not shown. Fig 6 uses the same data to rank all of the scenarios against each other, with color intensity indicating how much better the best option is over the second best. The former figure indicates how the strategies perform in absolute terms, while the latter indicates how dominant a strategy is as the pandemic and intervention proceeds.

In absolute terms (Fig 5), the most important factor is vaccine supply, with increasing supply leading to lower absolute infections and deaths. The trends in relative terms are more complex: in lower supply settings, there are generally more remaining outcomes to be averted compared to the baseline, so large relative gains can be made with more sophisticated strategies. Essentially, each dose if used appropriately can have a proportionally larger effect, because there is more to be prevented. However, there are fewer doses available to make those gains, so the absolute impact is modest. This leads to clearer separation in relative performance, compared to the absolute perspective, and critically makes it easier to see strategy impact differences between cumulative infections and deaths.

For infections, all of the scenarios begin similarly, but incidence diverges rapidly in 2021. In the HS setting, overall coverage is sufficient to blunt infections during the delta wave. Immunological resistance to omicron infection, to the extent that it existed, tended to come from delta infections in LS and MS scenarios, and from vaccination in HS and USA scenarios (top and middle rows of Fig G in S2 Text, respectively). Despite the significant discrepancies in

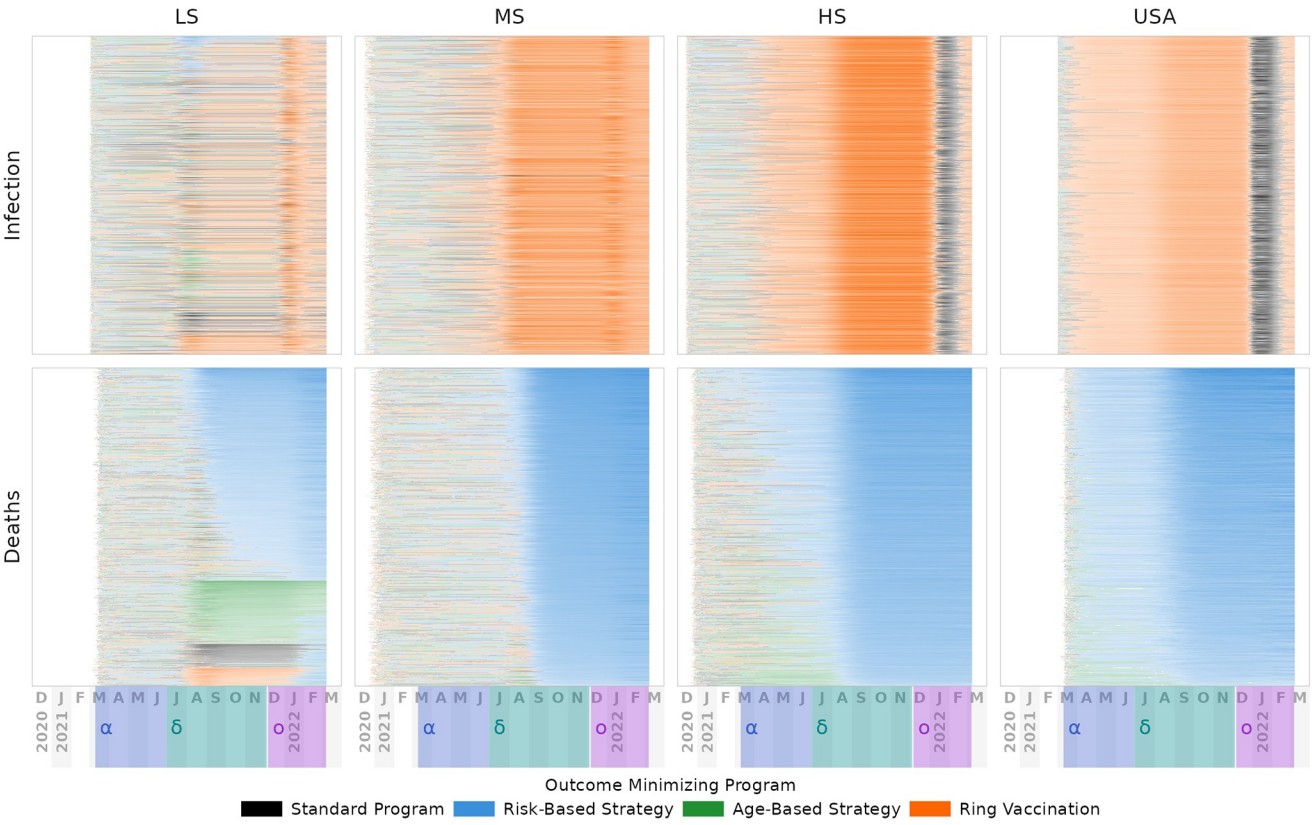

**Fig 6. Non-Quarantining Strategy Dominance.** As in Fig 5, columns represent vaccine supply scenarios and rows represent disease outcomes. Each row of pixels in a panel corresponds to one simulation replicate. Color indicates which strategy would prevent the most cumulative incidence had it been pursued up to that day. Replicates are ordered first by which strategy has the highest advantage in the most replicates, then by the magnitude of that advantage. Advantage is calculated each day as the additional prevented deaths versus the next-best strategy, and the overall advantage of a strategy within a replicate is the sum across all days. Relatively low advantage is mostly transparent in this figure, high advantage is mostly opaque. Strain dominance is indicated in the date band at bottom for alpha, delta, and omicron variants. Against infections (top row), the ring-based approach is typically preferable over the time period considered, with a brief interval during omicron where random vaccine distribution would have been preferred for high-supply scenarios. For deaths, the risk-prioritized strategy generally dominates within a few months of introduction, though for low-supply settings other strategies are preferred in a minority of replicates. See Figs K-M in S2 Text for strategies with quarantining, as well as without any seasonal forcing, but in summary: the same qualitative trends are present, *i.e.*, the vaccine distribution strategy preferences suggested by the model are independent of both quarantining and seasonality assumptions.

vaccine supply and epidemic curves, the mean risk of infection given exposure was similar across all vaccine supply levels (bottom row of Fig G in S2 Text) for the duration of the simulation. Because the model retains the complete infection and vaccination history for each person, it is possible to characterize the strain-specific extent and source of immunity throughout the simulated pandemic.

Quarantining of contact-traced individuals tends to dampen infections in the delta wave and earlier. Quarantine targets individuals most likely to be exposed (*e.g.*, workers), however, who tend to differ from those most likely to suffer severe symptoms (*e.g.*, seniors). When the omicron wave arrives, quarantining is not able to keep up with the increased transmissibility. Quarantine scenarios at that point have an immunity deficit compared to their no-quarantine counterparts, leading to a larger omicron wave.

However, the corresponding trends in deaths do not necessarily track with infections, as illustrated by comparing the top and bottom rows within Figs 5 and 6. In general, the programs that somehow prioritize disease-risk (*i.e.*, fully-risk-aware and age-based strategies)

generally prevent fewer overall infections than even random distribution of the vaccine. Yet, they are also the most effective at preventing severe disease outcomes including death. Conversely, the ring vaccination program has the effect of targeting people with many contacts, such as students and working age adults. As a result, ring vaccination tends to unintentionally prioritize people who are generally younger and healthier than average, and thus at lower risk of severe disease (see Figs B and O in S2 Text). Quarantining provided a consistent, if modest reduction in incidence of death—on the order of 10 to 15%—across supply levels and most vaccine programs, with the exception of ring vaccination in HS and USA simulations, where quarantine had negative cumulative effectiveness after omicron.

Apparent performance of a strategy depends on the timing of the assessment. We specifically considered comparison during periods of low transmission after epidemic waves as likely points of policy assessment. Fig 7 highlights trends in the simulation results, measured after each VOC wave. Generally, quarantining increases effectiveness regardless of vaccine supply, vaccination strategy, or VOC. Against infections, ring vaccination outperforms other vaccination strategies, though the effect is sometimes small and depends on both vaccine supply and assessment timing. Against deaths, a consistent vaccination strategy ranking (from best to worst performance) emerges regardless of vaccine supply or VOC: risk-based, age-based, standard, and ring vaccination. The discrepancy between most and least effective distribution

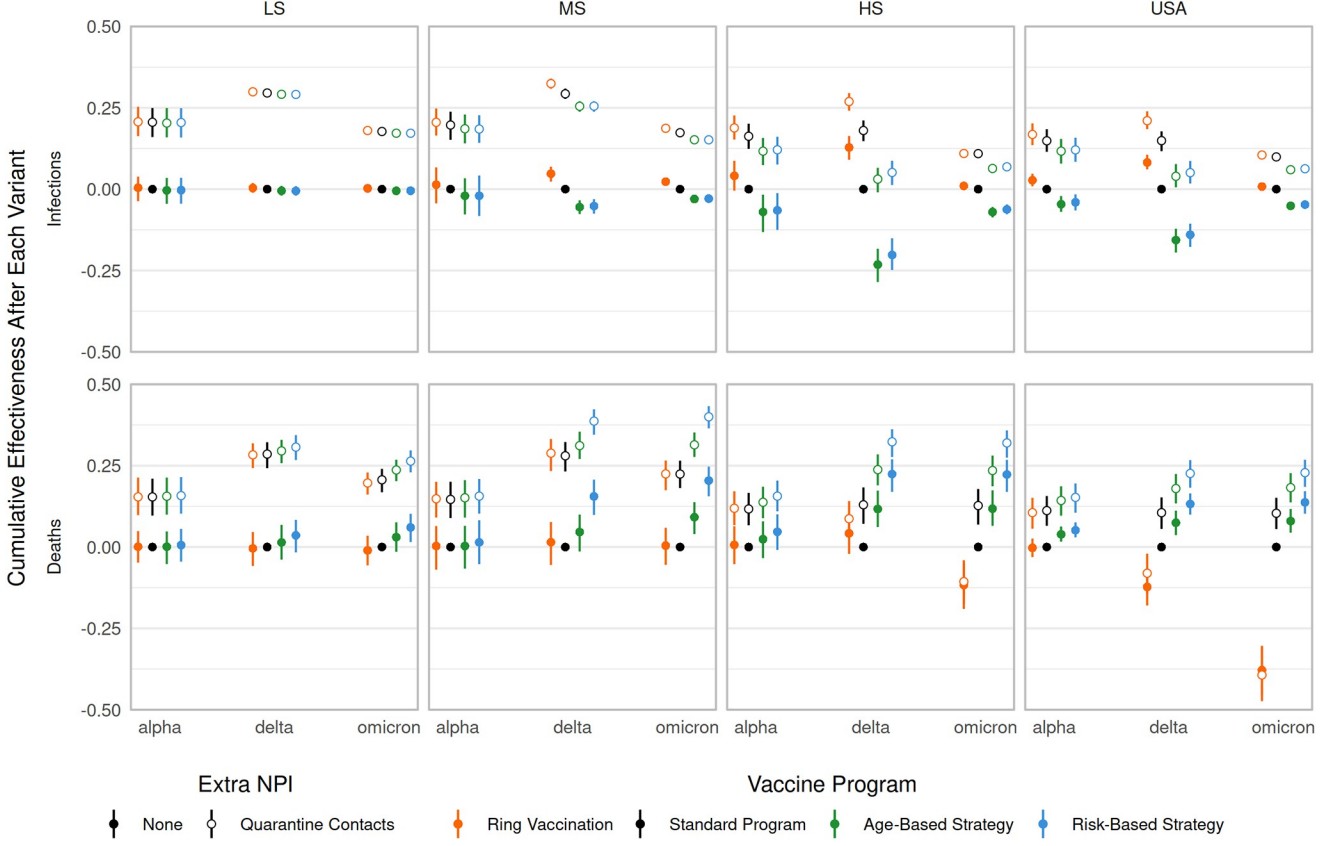

**Fig 7. Cumulative effectiveness after variant waves.** Columns depict vaccine supply scenarios and rows separate infection and death results; these are effectively "snapshot" values of the effectiveness curves at 2021-05-07 (post alpha), 2021-11-26 (post delta), and 2022-03-07 (post omicron) (see Figs J and N in S2 Text). "Waves" are defined generally as the time from when a VOC is introduced to when a new VOC is introduced (however the alpha period starts at the beginning of the simulation and omicron period ends at the end of the simulation). The non-quarantining, standard strategy is used as the baseline for all comparisons.

strategy tends to increase with vaccine supply. Finally, for both infections and deaths, with and without quarantine and across all supply levels, the cumulative effectiveness of ring vaccination decreased as a result of the omicron wave. This is likely due to both the VOC spreading more rapidly (via increased infectiousness and decreased latent period), and decreased efficiency of the distribution strategy, as a smaller and smaller fraction of those traced would be as-yet unvaccinated and thus eligible for vaccination.

Conditional vaccination, where vaccines are only given to individuals with no known infection history, provided modest additional benefits (compared to the main text, unconditional vaccination results) against death for LS and MS scenarios, particularly for risk-based vaccination (Figs H-J in S2 Text).

## Discussion

Using a spatially-explicit, stochastic agent-based model, we were able to calibrate transmission parameters to reasonably match a wide-variety of outcomes (including reported cases, hospitalizations, deaths, seroprevalence, and breakthrough infections) for the COVID-19 pandemic in the state of Florida from February 2020 to March 2022. Using that calibrated model and population, we evaluated vaccination strategies that cover both the actual COVID-19 response programs and several alternatives. We evaluated overall strategy performance, *i.e.*, at the scale of the entire population including non-vaccinees, in terms of infections and deaths. Strategies were compared to a "null strategy" standard program, where doses are administered randomly among the entire eligible population. We found consistent benefit from risk-prioritization programs for severe disease outcomes (*i.e.*, disease warranting in-patient medical care and up to and including death), across supply levels, with and without additional quarantining NPIs. Strategy performance against infections was more nuanced, with program ranking depending on both socioeconomic setting and the timing of the assessment.

When only considering overall infections and not deaths, the risk-based programs tended to rank lower in cumulative effectiveness than alternatives during the alpha and delta SARS-CoV-2 waves, but the resulting increased immunity from those infections led to reduced omicron transmission, minimizing the differences between strategies by the end of the omicron wave (see Figs 6 and G and K through M in S2 Text). On the other hand, ring vaccination strategies, which specifically target segments of the population where transmission is occurring, demonstrated the reverse pattern, preventing infections due to earlier variants but losing ground against omicron. Notably, while ring-vaccination generally prevented the most infections, it is largely the same as the standard program for severe outcomes, for all supply levels, unless supplemented with quarantining (see Fig 7). Because ring vaccination targets high-transmission risk individuals, and they tend to be young and working age adults, this approach has the effect of shifting the limited vaccine supply away from relatively older individuals at increased risk of severe outcomes. We expect that real-world use of ring vaccination for COVID-19 would be even less effective than our model's predictions, particularly during peaks in transmission, as we did not assume resource limitations when contact tracing, nor delays in vaccinating traced individuals. Realistically, contact tracing capacity is specific to both locale and methodology, and whether ring vaccination would be practicable during *e.g.* an omicron-like wave would need to be evaluated given those specifics.

Increasing vaccine supply from the LS to MS to HS levels provided increasing benefit to all of the distribution strategies. Increasing vaccine supply from LS to MS levels reduced cumulative deaths by approximately 32% regardless of quarantining or vaccination strategy, and further increasing supply to HS levels approximately reduced deaths by an additional 49%. However, these values assume an HIC-like population and infection-fatality rate [23]; the

absolute number averted would change with more context-specific assumptions, but the overall impact on relative changes is not obvious.

Against cumulative infections, the impact was somewhat less dramatic, largely because of the vaccine's modest efficacy against infection and infectiousness. Increasing supply from LS to MS levels reduced cumulative infections by only 9%, while increasing to HS levels reduced infections by an additional 25%. HS and USA scenarios had similar cumulative vaccine coverage, and provided similar benefit for these relative reductions in infections and deaths.

For a given vaccine distribution strategy, adding quarantining as an NPI provided only modest additional benefit at HS and USA vaccine levels (<10% reduction in infections and deaths). For lower supply levels, the impact was somewhat higher (approximately 10 to 15%). These are cumulative endpoints, measured after the omicron BA.1 wave. During the delta wave, we generally observed a bigger impact for quarantine, but quarantine was less effective against the more transmissible omicron variant. At the population scale, some of the infections that were avoided during delta were simply put off until omicron (see Fig E in S2 Text). While the benefits of quarantine suggest a potentially attractive NPI strategy, that decision would need to account for costs (which might be worse in settings with lesser access to *e.g.*, work-from-home alternatives) and transient threshold effects like running out of ICU beds, which we do not consider in this analysis.

Our approach has limitations, particularly when considering extrapolation to low and middle income countries, and generalization from the *average* supply levels to specific locales. We fit model parameters to data from a particular HS setting that (in comparison to LS and MS settings) is likely older and differs in comorbidities, economic and schooling activity patterns, and access to healthcare, among other distinctions. Using a detailed agent-based model makes it possible to evaluate equally detailed scenarios for other locales, but tailoring to specific settings requires both a large, varied collection of high-resolution non-epidemiological empirical data to construct the population (*e.g.*, household structure, business distribution, schools structure) as well as detailed epidemiological time series. Our quantitative results for incidence of infections and deaths should not be interpreted as estimates for particular income settings, as we do not have the relevant empirical data to calibrate the model, and countries vary widely within those categories. Relative strategy impact, *i.e.*, effectiveness, should be more reliable than incidence projections, as prediction errors will tend to have similar direction and magnitude across scenarios for a given setting, and the ordered ranking of approaches should be more reliable still. Also, while seasonality almost certainly varies by locale [24], perhaps most strongly by latitude, this does not seem to affect our findings. We performed a model fit and scenario analysis without any seasonal forcing and found similar quantitative results and the same program ranking, so while the specific seasonal model may not generalize to other regions, that does not seem to be an important consideration for strategy design.

The rankings of strategies are also pathogen- and, to a lesser extent, vaccine-specific. As such, future analyses of pandemic disease will need to repeat studies like this one to weigh the relative merits of different distribution programs. COVID-19 severity increases dramatically with age. This in turn increases the benefits of strategies that prioritize disease-risk, particularly in settings with relatively flat demographic distributions. Age is also a particularly easy, practical criterion to develop a strategy around. This severity profile is not a feature of all respiratory pathogens, however, let alone of pathogens generally. Additionally, SARS-CoV-2 is highly transmittable even in the absence of symptoms, which works against active response measures. Again, this feature is not ubiquitous among pathogens: even SARS-CoV-1 had a different-enough transmission profile for ring-based approaches to be effective.

Additionally, our model of risk perception and personal protective behaviors is essentially accounting for model residuals. When comparing the trends in inferred perceived societal risk

to epidemic trends in our reference population, we see perceived risk slightly lagging incidence dynamics, with gradually dampening reactions. This matches our intuitions about how populations actually reacted to pandemic waves, but this argument is largely anecdotal. Early in the pandemic, mobility data appeared sufficient to characterize behavior shifts, but we found this relationship diverged within a few months, plausibly due to the myriad social and business adaptations that occurred. More empirical studies are needed to identify what additional measures should inform transmission parameters beyond changes in mobility trends. Such work would both reduce the need for *ad hoc* residual adjustments and improve the portability of a behavior model. However, even these additional data would not necessarily address how behavior changes with prevailing pandemic conditions. That remains an important open question in infectious disease modeling, which needs focused work to inform what data to collect to appropriately parameterize explicit reaction models. We did not attempt to codify our approach to the behavior model in a way that would respond to prevailing conditions. Instead, we simply use the risk curves from the fitting stage for the alternative strategy scenarios. Those scenarios result in different incidence, which in real populations would likely affect risk perception and thus behavior.

Overall, our model indicates that disease-risk prioritizing strategies consistently generated greater public health benefit than mass or ring vaccination. This finding did not vary with distribution setting, the addition of quarantining, or by timing of the measurement. The most demanding scenario in terms of information on disease risk provided the best results, though generally the less information-intensive age-prioritization appears to provide a sufficient surrogate for disease-risk. The actual best policy choice would incorporate other factors, for example vaccine distribution cost and political feasibility, which while not incorporated in this analysis, we would expect to favor simpler programs.

## Supporting information

**S1 Text. COVID-ABM Description.**
(PDF)

**S2 Text. Additional Results.**
(PDF)

## Acknowledgments

The National Corporation Directory provided data on Florida businesses including their locations and NAICS categories.

The authors acknowledge University of Florida Research Computing for providing computational resources and support that have contributed to the research results reported in this publication.

## Disclaimer

The identification of any specific commercial products is for the purpose of specifying a protocol, and does not imply a recommendation or endorsement by the National Institute of Standards and Technology.

## Author Contributions

**Conceptualization:** Thomas J. Hladish, Alexander N. Pillai, Carl A. B. Pearson, Kok Ben Toh, Andrea C. Tamayo, Arlin Stoltzfus, Ira M. Longini.

**Data curation:** Thomas J. Hladish, Alexander N. Pillai, Carl A. B. Pearson, Kok Ben Toh.

**Formal analysis:** Thomas J. Hladish, Alexander N. Pillai, Carl A. B. Pearson, Kok Ben Toh.

**Funding acquisition:** Thomas J. Hladish, Carl A. B. Pearson, Kok Ben Toh, Andrea C. Tamayo, Arlin Stoltzfus, Ira M. Longini.

**Investigation:** Thomas J. Hladish, Alexander N. Pillai, Carl A. B. Pearson, Kok Ben Toh, Andrea C. Tamayo, Arlin Stoltzfus, Ira M. Longini.

**Methodology:** Thomas J. Hladish, Alexander N. Pillai, Carl A. B. Pearson, Kok Ben Toh, Andrea C. Tamayo, Arlin Stoltzfus.

**Project administration:** Thomas J. Hladish.

**Resources:** Ira M. Longini.

**Software:** Thomas J. Hladish, Alexander N. Pillai, Carl A. B. Pearson, Kok Ben Toh, Andrea C. Tamayo, Arlin Stoltzfus.

**Supervision:** Thomas J. Hladish, Carl A. B. Pearson, Andrea C. Tamayo, Arlin Stoltzfus, Ira M. Longini.

**Validation:** Thomas J. Hladish, Alexander N. Pillai, Carl A. B. Pearson, Kok Ben Toh.

**Visualization:** Thomas J. Hladish, Alexander N. Pillai, Carl A. B. Pearson, Kok Ben Toh.

**Writing – original draft:** Thomas J. Hladish, Alexander N. Pillai, Carl A. B. Pearson, Kok Ben Toh, Andrea C. Tamayo, Arlin Stoltzfus.

**Writing – review & editing:** Thomas J. Hladish, Alexander N. Pillai, Carl A. B. Pearson, Kok Ben Toh, Andrea C. Tamayo, Arlin Stoltzfus, Ira M. Longini.

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
