## [Decision Letter · Decision Letter 0]

24 Apr 2023

Dear Dr Hladish,

Thank you very much for submitting your manuscript "Evaluating targeted COVID-19 vaccination strategies with agent-based modeling" for consideration at PLOS Computational Biology.

As with all papers reviewed by the journal, your manuscript was reviewed by members of the editorial board and by several independent reviewers. In light of the reviews (below this email), we would like to invite the resubmission of a significantly-revised version that takes into account the reviewers' comments.

We cannot make any decision about publication until we have seen the revised manuscript and your response to the reviewers' comments. Your revised manuscript is also likely to be sent to reviewers for further evaluation.

Sincerely,

Alex Perkins

Academic Editor

PLOS Computational Biology

Virginia Pitzer

Section Editor

PLOS Computational Biology

Reviewer's Responses to Questions

**Comments to the Authors:**

Reviewer #1: Please see attached PDF.

Reviewer #2: The authors present a detailed, geospatial agent-based model of COVID-19 calibrated to Florida's epidemic and demographics, and use it to evaluate different vaccination coverage levels and strategies, finding that Florida's prioritization approach was close to optimal.

The manuscript is clear, and the model seems well designed and well documented. The following comments aim to clarify aspects of the methodology and results.

Manuscript:

- Fig. 3: I'm not sure mixing linear and log scales is especially helpful here...I was greatly confused why the omicron peak wasn't larger until I noticed this (and if the y-axis labels could be made darker, that would help).

- Fig. 3: The fit is impressive. That said, I'd be curious to know if seasonal forcing was really required to get a good fit (no need to rerun the calibration though, I'm sure it was an extremely involved process!).

- p. 9: Given the wide variability of vaccination rates between countries (https://www.thenationshealth.org/content/51/8/1.3), and especially between lower- and upper-middle income countries (https://data.undp.org/vaccine-equity/), the authors are encouraged to use specific country archetypes rather than lumping all LICs, MICs, etc.

- p. 12: I don't see anything about the delay between the vaccination event and the immune response. In practice this takes considerable time (days to weeks). While it is less important on a population level, wouldn't this delay need to be taken into account in order to have accurate ring vaccination results?

- Fig. 5: The labels are very confusing; it's easy (for me) to forget that "LIC" refers *only* to the amount of vaccine provided. Less "loaded" labels such as "Low vaccine coverage" are probably more appropriate.

- Fig. 5: The different strategies are too similar to tell apart on this graph. Why not make this into two graphs, one for different vaccine coverage levels and the standard program, the other for the different strategies for USA vaccine coverage?

- p. 20: "Our quantitative results for incidence of infections and deaths should not be interpreted as estimates for the LIC and MIC settings" -- I agree, to the point that I think these labels are doing more harm than good, even if qualified in the discussion as they are here.

- p. 21: "we are comfortable with this approach and suspect that attempts to compare to relevant empirical data (e.g., risk surveys, changes in mobility patterns) would be consistent." -- it seems odd that the authors would speculate on this when mobility data is available for Florida (Facebook, Google, SafeGraph, etc). Why not actually do this comparison? One would not expect there to be a perfect correlation, but if there's no correlation, that would suggest, at minimum, an alternative interpretation to "societal risk perception" as an overall transmissibility modifier.

Method supplement:

- The authors are thanked for their comprehensive supplement and open-source code.

- p. 1: How can there be 33,800 workplaces and only 118 schools? This implies there are almost 300 workplaces for every school, which doesn't seem right.

- It sounds like the model doesn't include waning immunity, except for the 60-day box function. This is unlikely to capture realistic immune dynamics, which show continuous, long-term exponential decay.

- Table S5: Are these efficacies, or risk ratios...? Does an efficacy of 1.0 imply perfect protection or no protection? I assume the former because they increase from doses 1-3, but doesn't VE_H = 1.0 imply that no one who is doubly vaccinated would die? I am confused.

- Table S6: Where do these values come from?

- Eq. S5: Why is viral shedding discretized into high and low shedding instead of being drawn from a distribution? Does the calculation of the Pareto distribution of infections take into account the fact that some people have many more contacts than others? (i.e. superspreaders both shed a lot and have above-average numbers of contacts.)

- Table S7: Why would there not be a fixed probability of contact rather than a fixed number? It doesn't make sense to draw more contacts in a workplace, say, than there are people in that workplace. Where does the assumption of 70% come from?

**Have the authors made all data and (if applicable) computational code underlying the findings in their manuscript fully available?**

Reviewer #1: Yes

Reviewer #2: Yes

PLOS authors have the option to publish the peer review history of their article (what does this mean?). If published, this will include your full peer review and any attached files.

Reviewer #1: No

Reviewer #2: **Yes: **Cliff Kerr
---

## [Decision Letter · Decision Letter 1]

20 Sep 2023

Dear Dr Hladish,

Thank you very much for submitting your manuscript "Evaluating targeted COVID-19 vaccination strategies with agent-based modeling" for consideration at PLOS Computational Biology.

As with all papers reviewed by the journal, your manuscript was reviewed by members of the editorial board and by several independent reviewers. In light of the reviews (below this email), we would like to invite the resubmission of a significantly-revised version that takes into account the reviewers' comments.

In particular, one of the reviewers identified a number of issues that need to be addressed for a positive decision to be possible. If these comments cannot be addressed satisfactorily in this round of revision, we will likely need to reject the manuscript at that point.

We cannot make any decision about publication until we have seen the revised manuscript and your response to the reviewers' comments. Your revised manuscript is also likely to be sent to reviewers for further evaluation.

Sincerely,

Alex Perkins

Academic Editor

PLOS Computational Biology

Virginia Pitzer

Section Editor

PLOS Computational Biology

Reviewer's Responses to Questions

**Comments to the Authors:**

Reviewer #2: The authors are thanked for their revision. While the manuscript is improved, having taken a closer look at it, I now have several additional methodological concerns.

1. Seasonal forcing: I still have concerns here (and the paper the authors cite isn't on SARS-CoV-2; it's from very early in the pandemic, extraploating from other coronaviruses). To my knowledge, no study has found conclusive proof of seasonality, although several studies have been suggestive. What data does exist seems to suggest a 12-month seasonal forcing term: see e.g. https://www.medrxiv.org/content/10.1101/2022.01.26.22269905v2.full-text. So now I am especially confused why the authors chose a 6-month term here. I can't find any explanation of the seasonality parameter in either the main text or the supplement. This aspect of the model either needs much more explanation and justification, or should be removed.

2. LIC/LMIC: The authors are thanked for making this change, but are asked to remove all remaining references to income levels (such as in the abstract). According to Our World in Data (https://ourworldindata.org/covid-vaccinations), the top 5 countries for vaccine doses per 100 people are Cuba, Chile, Japan, Brunei, and Cambodia. Needless to say, those do not fit into a neat income classification!

3. Fig. 5: Another way of phrasing my original comment is that there are 8 panels, each showing 8 scenarios: so 64 time series in total. This is fine for a supplement but seems like a lot for a paper. I want to ask: what's the bottom line? If the bottom line is, as the authors state, "strategy doesn't matter that much", there are probably simpler ways of making that point! I would much prefer seeing a handful of time series, and then a scalar metric (e.g. bar plot of final time points) for the full 8x4x2 matrix.

4. Mobility data: The additional clarification is appreciated. However, looking more closely at the "societal risk perception", I am having a hard time reconciling it with my experience. Especially in a state like Florida, I would've expected minimal changes in behavior after the initial wave of COVID. This is backed up by the mobility data (see e.g. https://ourworldindata.org/covid-google-mobility-trends) and face mask data (https://bmcpublichealth.biomedcentral.com/articles/10.1186/s12889-021-12175-9). In contrast, the "societal risk perception" seems to fluctuate from 0.5 to near-zero almost exactly every 4 months. I didn't quite follow Sec 5.4 of the supplement, but it sounds like the proportion of households that use "PPBs" are directly proportional to the current "societal risk perception", which would imply large month-to-month fluctuations in mask use. This doesn't seem to be supported by the data. The authors refer to "societal risk perception" as a "residual" in the model, and this seems like a potentially more correct explanation -- that there is a structural issue with the model that requires each wave to be artificially dampened. Perhaps mixing is too homogeneous? Perhaps immunity is waning too quickly? The authors are strongly encouraged to revisit their model assumptions and see if the magnitude of this "residual" can be reduced.

5. Workplaces: The issue is with equating "corporation" with "workplace". This is simply not valid. The extreme example is Delaware, with roughly two corporations for every adult and child in the entire state (1.9m corporations vs 1.0m people; see https://corp.delaware.gov/stats/). Removing "non-patronized" workplaces would go a long way towards addressing this (although ~30 workplaces per school still seems like too many workplaces to me!). What are the transmission implications of a workplace with a single employee -- none, right?

6. Vaccine efficacy: The citation given (https://www.medrxiv.org/content/10.1101/2021.05.20.21257461v2.full.pdf) was published only a couple months after the vaccines first became available. There are much better sources available now. More recent estimates put the efficacy at something closer to 60% (https://jamanetwork.com/journals/jamanetworkopen/fullarticle/2802473). It is essential to update this parameter, as the findings will likely change significantly.

**Have the authors made all data and (if applicable) computational code underlying the findings in their manuscript fully available?**

Reviewer #2: Yes

PLOS authors have the option to publish the peer review history of their article (what does this mean?). If published, this will include your full peer review and any attached files.

Reviewer #2: No
---

## [Decision Letter · Decision Letter 2]

2 May 2024

Dear Dr Hladish,

We are pleased to inform you that your manuscript 'Evaluating targeted COVID-19 vaccination strategies with agent-based modeling' has been provisionally accepted for publication in PLOS Computational Biology.

Best regards,

Virginia E. Pitzer, Sc.D.

Section Editor

PLOS Computational Biology

Virginia Pitzer

Section Editor

PLOS Computational Biology

Note that Reviewer 2 expressed additional concerns about the seasonality assumptions; however, I feel this has been adequately addressed by the sensitivity analysis. Nevertheless, you may wish to provided additional justification in the Methods section prior to publication.

Reviewer's Responses to Questions

**Comments to the Authors:**

Reviewer #2: The authors are thanked for their response, especially clarifying vaccine efficacy. While most of the comments have been addressed, I am unfortunately not convinced regarding the seasonality and "societal risk perception". To me, rather than assume seasonality varies ±15% with peaks in January and July, and assume all additional changes in force of infection are due to "societal risk perception", I believe it would be more scientifically valid to acknowledge that these factors (and others) cannot be disambiguated. In practice, this would involve multiplying the time series shown in Fig. 3(a) and Fig. 3(d) as an overall modulation of force of infection (without, if I understand it correctly, actually changing the model results), and let the reader come to their own conclusions about whether it maps onto seasonality, risk perception, or something else.

**Have the authors made all data and (if applicable) computational code underlying the findings in their manuscript fully available?**

Reviewer #2: Yes

PLOS authors have the option to publish the peer review history of their article (what does this mean?). If published, this will include your full peer review and any attached files.

Reviewer #2: No

---

## [Editor Report · Acceptance letter]

29 May 2024

PCOMPBIOL-D-23-00170R2 

Evaluating targeted COVID-19 vaccination strategies with agent-based modeling

Dear Dr Hladish,

I am pleased to inform you that your manuscript has been formally accepted for publication in PLOS Computational Biology. Your manuscript is now with our production department and you will be notified of the publication date in due course.

With kind regards,

Zsofia Freund
